# Cultivating the Acceptance of Assistance Dogs in Aged Care through Deliberative Democracy

**DOI:** 10.3390/ani13162680

**Published:** 2023-08-20

**Authors:** Amanda J. Salmon, Nancy A. Pachana

**Affiliations:** School of Psychology, University of Queensland, Brisbane, QLD 4072, Australia; a.salmon@uq.edu.au

**Keywords:** assistance dogs, aged care, older adults, regulation, support

## Abstract

**Simple Summary:**

The use of assistance dogs has the potential to provide a myriad of benefits to older adult owners, far beyond mere utility. However, despite legislation designed to prevent discrimination in accommodation for owners and assistance dogs, many aged care facilities continue to not allow owners to retain their dogs on relocation. Through deliberative democracy, the study used a panel of key stakeholders to explore the issue. Specifically, the complexities behind whether owners should be allowed to retain their dog, what should be considered in making this decision, and what the best practice would be in allowing this. It was suggested that by introducing objective initial and ongoing assessments for the owner, dog, and facility, it would allow for a fair decision that considers the safety and well-being of all involved. Further, the implementation of sufficient policies and procedures would help support all involved, whether the dog is able to be retained by the owner or not.

**Abstract:**

Assistance dogs provide significant benefits to older adult owners. However, despite protective legislation, aged care facilities continue to not allow owners to retain their dogs on relocation. The purpose of the current study was to explore whether older adults should be allowed to retain their dog on relocation to an aged care facility, and what factors should impact this decision. Further, if allowed to retain their dog, what would be the best practice to allow for this? A deliberative democracy methodology was used, with a range of key stakeholders recruited. Focus groups were held, with follow-up questionnaires to establish deliberation for all questions. Results indicated that with sufficient objective measurement, fair decisions can be made to ensure the welfare and well-being of the owner and dog. Key policy and procedure changes would also be necessary to ensure ongoing support, such as training, care plans, and emergency directives. By ensuring sufficient policies and procedures are in place, training and support could lead to an ideal outcome where facilities could be at the forefront of a better future for aged care.

## 1. Introduction

The use of assistance dogs can have a profoundly positive impact on the lives of their owners, with considerable focus on the recent research body. This impact is especially crucial for older adults who are also more likely to experience health declines, such as in vision and hearing, which warrant the use of an assistance dog. Further, it is important to note the benefits extend far beyond mere practical tasks, but also across mental, physical, and social health domains [1]. As such, it seems rational that older adults should be encouraged to acquire an assistance dog as a multi-faceted treatment and support. However, a number of actual and perceived barriers prevent or deter this age group from this acquisition, in particular the availability of suitable accommodation.

Current legislation, at state and federal levels, including the Guide, Hearing and Assistance Dogs Act 2009 [2] and Disability Discrimination Act 1992 [3], is designed to protect the rights of owners and their assistance dogs. This includes maintaining their right to sufficient accommodation, outlawing discrimination based on refusal as well as any condition that would lead to their separation. Despite this, it has been suggested that many aged care facilities in Australia continue to refuse to allow owners to relocate to the facility with their assistance dog, with one report finding as little as 18% allowing residents to keep an animal of any kind [4,5]. This is a considerable barrier not only in discouraging initial acquisition, but in their retention. The latter potentially contributes to significant distress for both owner and dog, with the loss or separation from an assistance dog creating distress greater than that of a companion dog [6].

The “One Health” initiative posits that the welfare of humans, animals and their environment are interconnected, and thus works to optimise overall community health by taking a collaborative approach implicating professionals across the human, animal, and environmental domains [7]. The “One Health, One Welfare” framework for human–animal interactions extends on this by using the same collaborative approach to achieve health and well-being benefits [8]. This focuses not only on the reciprocal human–animal relationship but the effects of the environmental and social factors in which that relationship exists. When applied to the current study, it can be seen that this is an issue with the potential to affect a number of key stakeholders, and it is thus important to consider the issue through the lens of a wide range of people who are either directly or indirectly impacted, such as allied health staff, animal professionals, aged care staff, and assistance dog owners themselves. These individuals are likely to have competing interests and differing opinions, all of which should be harnessed for well-rounded discussion and solutions. Deliberative democracy involves providing participants with adequate information, before facilitating a discussion that not only takes into consideration differing views but works to integrate the views of all participants based on a culmination of all participants’ perspectives [9]. This methodology has received support for use within the public policy sphere, with the opportunity to raise and consider alternative perspectives increasing the likelihood of policy acceptability and thus increased likelihood of successful implementation [10]. While traditionally these discussions would be held in person, the boom of online meeting software has allowed for the adaptation of online deliberative democracy. While this comes with possible technical difficulties, it has been found to lead to the same outcomes [11], with the benefits of improved scheduling and recording capabilities, and the ability to complete follow-ups with participants via email. This allows for increased capabilities to reach consensus on key issues.

The current study thus aims to use deliberative democracy to explore whether older adults with assistance dogs should be allowed to retain their dog when they relocate to an aged care facility, and what factors should impact this decision (e.g., dog size, care abilities). Further, if they were allowed to retain their dog, what would be the best practice to allow for an effective transition and continued support within the aged care facility?

## 2. Materials and Methods

### 2.1. Participants

Participants (N = 18) were recruited via a convenient sample of existing professional networks. The authors selected professionals and consumers with a range of backgrounds and experiences, all of whom had relevant experience or involvement with older adults, aged care, or assistance animals (See “Table 1”). Potential participants were initially invited by direct email, and those who indicated interest were then sent participant information, consent form, and instructions to nominate their available dates and times via an online poll. The study was approved by the Research Ethics and Integrity Board at the University of Queensland (2022/HE001752), and informed consent was received from all participants prior to the commencement of the study.

### 2.2. Design

The current study used an adapted exploratory deliberative democracy methodology with qualitative and quantitative findings. Qualitative data were collected during focus groups and were summarised into key common points. Where deliberation of key points was not reached, or response priority was unclear, the points were developed into a questionnaire format for ranking, providing subsequent quantitative data.

### 2.3. Procedure

On receipt of consent forms and time availability from all participants, 3 online focus groups were held via Zoom and facilitated by the primary author, with numbers kept as even as possible to allow for minimal group sizes to promote discussion and engagement from all participants (Group 1, *n* = 6; Group 2, *n* = 6; Group 3, *n* = 6). While ideally, this allocation would consider the spread by participant experience, availability needed to take precedence due to participant availability.

All focus groups followed an identical format, facilitated by PowerPoint slides containing all crucial information. Firstly, participants introduced themselves and their relevant backgrounds. Secondly, they were presented with background information on the issue (See “Appendix A”) to ensure all had a base-level understanding of the issue. Thirdly, participants were provided with a list of key assumptions for the subsequent case studies, to ensure the brevity and specificity of discussion across all groups. These included having them assume that in both case studies the assistance dog owner was:Healthy enough to care for their dog themselves;Not having cognitive functioning impairments;Intending to relocate from their home to an aged care facility;Previously living alone.

Fourthly, they were presented with 2 differing case studies and a list of questions. This not only guided the conversation and ensured relevant discussion, but also gave tangible examples of how the cases may or may not differ within their considerations. These included:Case 1: Person A has a severe hearing impairment, for which she has a hearing dog to assist in alerting her to key sounds (e.g., the doorbell, kettle, and smoke alarm). She has owned her hearing dog for 5 years, which is a small terrier.Case 2: Person B is blind, and he has a guide dog to assist in his mobility. He previously used a cane but did not find this to be as effective. He has owned his guide dog for 5 years, which is a large Labrador.

Questions:What would an appropriate assistance dog policy for the aged care home look like?Should these policies differ across the two presented cases? Why/why not?Are there any other things that need to be implemented by aged care facilities to allow for owners to keep their assistance dog in the facility?Are there any relevant bodies that should be involved in ensuring these policies are being upheld?Is there any other information you would want to know about the cases that would affect your decision?

On completion of all focus groups, discussions were transcribed with identifying data removed. Data were summarised (See Section 2.4) and questions that did not have agreement had their various responses input into questionnaire format for ranking, using Qualtrics. These data were then collated and analysed through examination of the means and standard deviations from the Qualtrics data output.

### 2.4. Analysis

The initial focus group data were examined by the author, with key points identified and summarised, with similar or related responses combined for each question. For Questions 1 and 2, sufficient deliberation and agreement were achieved during the focus groups. Sufficient deliberation and agreement were achieved during the focus groups. In other words, the majority of participants (over 75%) agreed on the points raised, thus subsequent rounds of deliberation were not needed. For Question 3, two key subthemes were identified and included in the questionnaire as two independent answer lists for ranking by importance. Question 4 responses were input for ranking by importance with no subthemes. Finally, Question 5 had three key subthemes identified, which were input for independent ranking. Subsequent participant data from this questionnaire were then analysed using the Qualtrics survey results output to examine the results of the ranking. Ranking was from 1 (most important) to the highest number, depending on the number of responses to rank (least important). Overall order of importance for each question or sub-question was established through the mean ranking response. That is, the lowest ranking mean was considered the most important response, the second lowest the second most important, and so on.

## 3. Results

The following will provide a summary of the results attained by question, with focus group qualitative data only for Questions 1 and 2 where deliberation was reached, and ranking was not necessary. The remainder of the questions include both the qualitative and quantitative results, with deliberation then established.

### 3.1. Question 1—What Would an Appropriate Assistance Dog Policy for the Aged Care Home Look Like?

Five key subthemes were identified from the qualitative focus group data, including (1) Assessing and preparation for the future; (2) Where the owner has limited ability to care for the dog; (3) Where the owner is unable to care for the dog; (4) Consideration of others (staff, clients, visitors); and (5) Dog welfare and ability to complete tasks. The following are the key points identified for the development of appropriate policies for aged care facilities:

#### 3.1.1. Assessing and Preparation for the Future

Assessing the ability of the person to care for their dog: aged care facilities should implement an objective testing protocol for whether the person is able to sufficiently care for the dog prior to relocation to the facility. This may require the development of a new assessment, or the direct use or adaption of an existing measure, such as the Companion-Dog Multi-Species Risk Management Tool (CAMSRMT) [12] or Safe Dog-Friendly Eldercare (SAFE), which is currently under development [13]. This should also be re-administered at regular intervals (annually; biannually) or after a significant event, such as hospitalisation.


*At the University of South Australia Janette Young produced a risk tool for aged care facilities to allow them to consider pet animals. Something similar including assistance animals may be useful.*


Assessing the welfare of the dog: aged care facilities in conjunction with owners should implement an objective protocol for whether the facility is appropriate for the dog prior to relocation and whether the dog’s welfare is continued to be supported. This would assess the suitability of the facility of the space itself (e.g., access to toileting facilities), and whether the dog is coping and is sufficiently cared for (groomed, fed, seeing a vet regularly). Also, whether the dog is receiving sufficient stimulation and exercise to maintain health and prevent issues, such as obesity. This may require engagement by the issuing body to do initial and regular assessments. This should also be re-administered at regular intervals (annually, biannually) or after a significant event, such as hospitalisation.


*There needs to be an appropriate support or care network for the dog. The Guide Dogs Association or other association should put in place something so that the dog is looked after, otherwise it will be overweight, not groomed, and defecating where it shouldn’t—dogs needs should be thought of.*


General dog care plan: where someone is assessed as being able to keep their dog in the facility, there must be a care plan in place for relevant services including who and regularity (e.g., yearly veterinarian, monthly groomer), basic care (e.g., how food will be acquired), and any medication or other needs of the dog.


*There can be issues with toileting dog, grooming, family not coming in to take the dog to the vet, nobody bringing in dog food, al an issue with who should be responsible.*


Dog risk management and emergency directive: aged care facilities in conjunction with owners should establish sufficient protocols to manage who will take care of the dog in case of short- or long-term inability to care for the dog, such as periods of illness or hospitalisation, and where the dog should be placed in the case of death or incapacity of the person. This may include partners, friends, family, RSPCA, or the supplying organisation (e.g., Guide Dogs Qld).


*When it comes to policies, having any animal needs a contingency form stored with a care plan is a necessity. Knowing who to call, for example Guide Dogs Australia. Those without an accredited body also need to be considered. For the one day staff can look after the dog, within three days they need to confirm who will pick up the dog, and in days they need to collect.*



*Policies need to be able to cater for changes in health needs. While capable at this stage and able to care for dog, I imagine health will deteriorate so policies should be adjusted for that person’s time in the facility to adapt to changing needs.*


Policy perspective broadly: developed policies should include overarching policies for inclusion and risk assessment, but also include provision of case-by-case individual management and assessment for acceptance. These considerations should apply not just due to differences in the owner or dog, but to other factors such as room configuration, for example if the room is too small for a large dog but could accommodate a small dog.


*Legally there has to be accommodations made so they [the owner] have adequate ability to care for the dog and they [aged care staff] need to consider how they can accommodate the dog and owner safely.*



*From a policy perspective, it needs to be case by case, with overarching policies for inclusion and risk assessment, but case by case management and acceptance. For example, risk management plan, and placement and room config can be affected by dog size.*


#### 3.1.2. Where the Owner Has Limited Ability to Care for the Dog

The dog may still be able to be retained where a care plan can be made inclusive of additional help. This should specify what help is needed by the owner (feeding, walking, toileting, grooming, vet attendance) and who will provide this support (family/friends, or volunteers/staff with special training or experience). Other short-term assistance may also be specified in a care plan, such as foster carers or involvement of the dog issuing organisation for temporary dog respite or crisis situations.


*So, ensuring a human-care plan and an animal care plan… Develop a plan of what’s going to happen to the animal, including who will take the dog when they die, who is the vet, and foster care or short care arrangements for the animal. Animal needs must be considered too.*


#### 3.1.3. Where They Are Unable to Care for the Dog

Aged care facilities should establish protocols to transition the patient into relocating to the facility without their dog. This may include slowly decreasing time with the dog and decreasing their reliance, having their own dog visit where possible or having a visiting therapy or companion dog they can engage with.


*Maybe send the dog to be with a family member, but it can come in and visit the person as a transition, so they still see the dog, but don’t have the responsibility. For example, when they know they can’t care for the dog anymore, but they can visit instead and transition out. Give option to help find new home and give the option of visiting periodically until they die.*



*They [aged care] need a good pet therapy program to allow for transition where they [the owner] can’t have their animal. It allows them to still be involved, for example allowing them to brush the dog.*


#### 3.1.4. Consideration of Others (Staff, Clients, Visitors)

There is a clear need to consider the needs of staff, clients, and visitors, including dog allergies, fear or dislike of dogs, and ethnic consideration, such as cultural beliefs of dogs being “dirty”. This may require some form of separation (e.g., a child gate on the doorway of the patient’s room), signage warning others of the presence of a dog, and training protocols and provision of information such as brochures about assistance dogs.


*Basically, anyone not at risk, for example of falls, can be managed in existing policies about creating best practice, for example a notice on their door alerting that the person has a dog in the room to alert staff with phobias/allergies.*


There is also a need to ensure the dog is kept at a hygienic standard to be around others, including being groomed, bathed, and nails cut so as not to damage anyone’s skin and minimise any infection and illness risks. This should tie into the ongoing assessment of the owner’s ability to care for the dog and ensure the standard of the dog is maintained for its welfare.


*Consideration needed to be made for the welfare of [the] dogs. For example, if they are not well groomed, not bathed, and have long nails which is bad for those with delicate skin.*


Housing and restraint should be specified as part of OHS policies, such as: the dog must be on a leash when outside the person’s room, and how it will be restrained within the room where needed (e.g., when moving beds or wheelchairs).


*They [the owner] needs to consider how to house the animal when there’s a lot going on in the facility, like moving beds, and wheelchairs, and whether they should be on a lead to help with fear of dogs in others.*


There could be an option for nominated additional staff training to be able to assist or provide information as a “champion” for assistance dogs. This may also include specific identification, such as a paw emblem on their name tag.


*They [aged care] could have nominated staff training with the choice to complete the training and have it on their name tag like LGBT advocates and languages etcetera. So, they [aged care staff] could get a little paw or something and be recognised as someone where you can go to them for help with your animal.*


#### 3.1.5. Dog Welfare and Ability to Complete Tasks

Some dogs may be reactive to the environment (e.g., smells, sights, and sounds), or be not used to a lot of attention, causing unwanted behaviours (e.g., toileting on the floor) or may become distracted from their working tasks, or engage in risky behaviours (e.g., swallowing medication dropped on the floor). As such, the individual dog should be continually assessed for suitability and problem-solving should guide any necessary policy or procedure changes.


*Some dogs are not used to a lot of attention causing other behaviours like toileting on the floor. Some are dogs interested in what was going on, triggering other [unwanted] behaviours.*


Other visitors may bring their own dog or visiting dogs to the aged care facility, which could become disruptive to assistance dogs without the same level of control. This may entail policies around separating any visiting dogs and providing education to the visitors around assistance dogs.


*An aged care facility I worked at family pets could be brought in. Procedures were in place and they continually reminded people it was for a specific day only.*


Where the owner is ageing in place and is unable to go outside or engage in activities, there should be an assessment of the usefulness of the dog in terms of task completion. This should ideally be completed by the organisation that issues the dog or training and provides a discussion around whether the dog could be retrained (e.g., to take the owner to a dining room or react to alternative sounds) or if the dog should be retired or relocated.


*Under the Queensland Act there’s expiry dates so they [the dog] have to be at certain level to perform tasks…. I would be more inclined to say a guide would be more useful because tasks are taken away from a hearing dog to degree, for example no doorbells. They need at least 3 tasks for them to do.*


### 3.2. Question 2—Should These Policies Differ across the Two Presented Cases? Why/Why Not?

All group participants unanimously agreed that any policies should not differ across the two case studies. It was noted that this could inadvertently introduce other discrimination issues and decisions should be made on a case-by-case basis and as objectively as possible.


*Consistency is very important in this space. You don’t want to introduce different types of discrimination.*


### 3.3. Question 3—Are There Any Other Things That Need to Be Implemented by Aged Care Facilities to Allow for Owners to Keep Their Assistance Dog in the Facility?

For Question 3, two key subthemes were identified, including (1) the physical environment; and (2) staff or training. Each subtheme contained multiple responses which were presented to the participants for ranking by importance. The following responses are presented by most to least important by mean ranking. For a summary of response statistics for Question 3, see Table 2. An additional yes/no question was also included in this section (see Section 3.3.3) as during one of the focus groups a “pay-for-service” model was suggested as a suitable resolution for those requiring additional care. As such, it was deemed by the authors as a question worthy of addressing with the group of participants as a whole.

#### 3.3.1. The Physical Environment

First important response: New aged care facilities should be built with dog retainment in mind, and established facilities should consider retrofitting wherever possible (possible government grants). This may include sufficient access to the outdoors (e.g., a balcony with a dog door, access to larger dog-friendly outdoor space for exercise/toileting), ability to keep the dog within the room (particularly where the owner is not in the room, e.g., a child gate on the room’s door), and sufficient storage space for dog food and/or toys.


*Design of aged care. They [aged care] need more external space for residents. Some could have access to a small courtyard off room; adapt this to house people better. Bringing this to existing centres may be too difficult, but future facilities could accommodate this.*



*It can be done. We currently have a [dog] recipient who is in aged care in Tasmania. The only change made to lodgings was a baby gate at front of room, there was enough space, and she cares for dog and takes it out.*


Second important response: Where a facility wants to maintain separation for those with fears or allergies or has a sufficient number of dogs, it could consider having dog-specific floors/wards where and those that are dog free.


*Maybe for residents who don’t like animals you could have animal specific floors where there’s a floor with no animals, one with.*


Third important response: Rooms should be configured to allow staff movement, emergency response or movement of equipment while keeping the dog safe or restrained where necessary.


*Configuration of rooms [is important] to allow for an emergency response while keeping the animal safe. For example, getting a hoist in to help after fall but getting around the animal and the animal’s things.*


Fourth important response: Where facilities have sufficient difficulties in housing a dog that cannot be adjusted (e.g., narrow corridors or rooms are too small), it must be acknowledged that they simply cannot allow the person to retain their dog. This is not just discrimination but introduces other legal issues. Where this is the case, the owner should consider an alternative facility.


*Relocation [of the dog] may not be appropriate depending on the case. Not all dogs can sit in a small room. Some rooms may be too small or inappropriate for dogs.*



*The built environment is very important, for example narrow corridors, so it does mean in aged care some just would be ruled out to even consider animals. It’s not discrimination, but other legal issues.*


#### 3.3.2. Staff or Training

First important response: Facilities could engage dog care volunteers (e.g., vet students, community volunteers) to provide care for any dogs in the facility.


*I’m seeing links between universities with students volunteering, so no cost needed and it would be a great opportunity to have younger people in general from the Vet School getting experience, but also links between them and the residents.*


Second important response: Facilities could employ a person(s) at the facility specifically for dog care.


*Where my parents live it’s pay for use for many things, so if they needed someone to come walk them once a day, they could pay an extra fee to have that.*



*We need aged care to employ one or more people with the skills to care for the dog to ensure they get appropriate care and not compromise the care of the dog for the benefit of them being there.*


Third important response: Facilities could provide optional training for current staff to provide dog care.


*It’s a good idea to have people trained and comfortable in animal care for those who have a sudden health shock or something gone skew-if, so there’s someone there to ensure the dog is cared for and comfortable.*


#### 3.3.3. Additional Question

Should the provision of dog care support (e.g., walking, feeding) be a pay-for-use service in an aged care facility? (Yes/No). Responses were *n* = 11 (64.71%) for “Yes” and *n* = 6 (35.29%) for “No” (See Figure 1).

### 3.4. Question 4—Are There Any Relevant Bodies That Should Be Involved in Ensuring These Policies Are Being Upheld?

No subthemes were identified for this question, so all answers were presented for ranking by importance. For a summary of response statistics for Question 4, see Table 3.

First important response: Aged care bodies: (e.g., Aged and Community Care Providers Association (ACCPA) and Catholic Health Australia (CHA)), to assist in ensuring there are necessary plans and supports in place.


*We need to bring in peak bodies to keep them on board, for example ACCPA and CHA to ensure there are the necessary plans and supports in place.*


Second important response: Assistance dog organisations: e.g., Guide Dogs Qld and Lion Hearing Dogs, who may complete the regular assessments.


*Ideally, the issuing body needs to come out and do regular assessments, should be yearly, and assess the suitability, is the dog happy, is the dog coping and cared for.*


Third important response: Dog rights organisations: e.g., RSPCA, to ensure all facilities have advocacy resources and dog welfare is maintained.


*We need to ensure available resources are present for animal rights organisations like the RSPCA so all facilities have advocacy resources for people.*


### 3.5. Question 5—Is There Any Other Information You Would Want to Know about the Cases That Would Affect Your Decision?

For Question 5, three key subthemes were identified, including (1) the person; (2) the dog; and (3) the owner–dog relationship. Each subtheme contained multiple responses that were presented to the participants for ranking by importance. For a summary of response statistics for Question 5, see Table 4.

#### 3.5.1. The Person

First important response: How traumatic is it for people to give up their dog? Some people may really struggle, so they need to have conversation at the early stages when getting the dog and providing mental health support where needed on relinquishment.


*Some really struggle, so need to have conversation at the early stages when getting a dog. Time goes fast before retirement… Not everyone is a “dog person” it’s just a tool to some people. It’s traumatic when it’s not dealt with properly.*


Second important response: Why the person is relocating to an aged care facility. If it is due to difficulty caring for themselves or has become acutely unwell, they may experience difficulty caring for their dog.


*Some aged care facilities have legitimate reasons for why they shouldn’t allow the dog, for example cognitive impairments and risk of dying within 12 months is very high.*


Third important response: Any relevant factors around their visual/hearing abilities, including severity and comorbidities, and available alternatives (e.g., blind cane).


*Any comorbidities. They may be blind but have other issues happening too. For example, they could have dementia and be blind. This affects their ability to care for the animal, and can be compromised quickly.*



*How else are they managing their impairments? They may need other devices or strategies implemented or get others in place if losing their dog, or other environmental changes may be needed to support them.*


Fourth important response: Whether they have any family/friends who are nearby or visit who are able to assist them with the care of their dog, or to take the dog in crisis.


*Depending on family support—who else can take care of the dog?*


#### 3.5.2. The Dog

First important response: The overall manners and any problematic behaviours of the dog to assess suitability.


*Not about the size, so much as the manners and care of the dog, which can affect if it can live in a small place.*


Second important response: Is it self-trained or provided by an organisation? This may affect who is able to follow-up on dog welfare.


*Problems with owner trained dogs are an issue still… Policies needs to say there needs to be an organisation involved.*


Third important response: Breed specific issues—what are the different needs of the breed, e.g., exercise, socialisation, stimulation, behavioural anomalies, etc., to consider.


*Need to consider breed specific issues. What are the different needs, like exercise, socialisation, stimulation, behavioural anomalies, etcetera to consider?*


Fourth important response: Age of the dog—if they have had it for 5+ years it may be heading towards retirement. This may require a discussion with the dog providing organisation.


*What is the age of the dog? If they have had it for 5 years, it may be heading towards retirement. Most bodies placing assistance dogs will have a conversation at assessment... Animals may need to be prematurely retired, so this may be a good time to have that conversation.*


Fifth important response: Availability of someone to physically pick up the dog in an emergency, e.g., managing a Labrador if they become sick.


*Need to consider having to physically pick them up in an emergency, for example managing a Labrador if they are sick.*


#### 3.5.3. The Owner–Dog Relationship

First important response: What is the cost to them if they are separated? Consideration for transition time—if losing the dog will make them deteriorate quicker then it could have higher offset or needs.


*What is the cost to them if they are separated? Consideration needs to be made for transition time. If losing the dog will make them deteriorate quicker then they could have a higher offset of needs. How long is the dog useful with that person? When will it retire?*


Second important response: Anything in particular about the person and dog, such as the context of where they came from (e.g., from a house vs. apartment) and what they would need to relocate. This would allow transition into aged care to be tailored.


*Anything in particular about the person and dog. The context of where they came from and what they would need to relocate. [Aged care] should tailor the transition into aged care, everything a bit different about the person, dog and relationship.*


## 4. Discussion

The current study aimed to use deliberative democracy to explore whether older adults with assistance dogs should be allowed to retain their dog when they relocate to an aged care facility, and what factors should impact this decision (e.g., dog size, care abilities). Further, if they were allowed to retain their dog, what would be the best practice to allow for an effective transition and continued support within the aged care facility.

The question of whether older adults should be allowed to retain their dog when they relocate to an aged care facility is deceivingly complex. At first glance it seems like a simple yes, particularly given the numerous and far-reaching benefits highlighted in the research [1] and the legislation, which is designed to prevent separation in the context of accommodation provision [2,3]. However, the results indicated that there is much to consider from the perspective of the owner, the dog, and the aged care facility to protect the best interests, safety, and well-being of all involved. Further, this needs to be grounded in sufficient policies and procedures based on objective measurement to minimise discrimination or ageist assumptions. Specifically, it was found that assessment should begin prior to relocation and be an ongoing process, including whether the owner is able to care for the dog, whether the dog’s welfare is maintained, and initially, whether the aged care facility is sufficiently and safely able to house the dog. While this seems daunting, the discussion highlighted some tools, such as the CAMSRMT or SAFE [11,12], that can be easily implemented. Further, many aged care facilities already maintain care plans for residents, which could be adapted to include dog care. So, while this may take some time to initially establish, ongoing assessment and record keeping would be achievable with minimal time or financial burden to the aged care facility.

On consideration of what factors should impact the decision, all participants unanimously agreed that policies should not differ across the case studies. This was noted to prevent any other inadvertent discrimination and to encourage decisions made on a case-by-case basis guided by objectivity. However, it was later raised that there were a number of factors that would influence this decision from a person, dog, and owner–dog perspective. The most important personal factor was “how traumatic is it for people to give up their dog?”. While there may be some who see their dog as a tool that is no longer necessary, research suggests that this may indeed be traumatic [6] and could be established through a discussion with the owner. Other factors alluded to how unwell the person is, the severity of illnesses or disability, comorbidities, and availability of friends/family to assist. These are all important factors for all parties involved, which could be addressed by the aforementioned initial and ongoing assessments and having a sufficient care plan in place. It was also noted that alternatives could be considered, such as the cane. But taking this example, it has been found that those who have used a guide dog for a prolonged period may not be as proficient with a cane due to lack of practice so this may not be easily implemented [14]. Considerations around the dog, such as manners and behaviours, training, and breed issues should also be assessed for suitability. While ideally factors such as dog size should not prevent an owner from relocating with their dog, it must be considered that some facilities simply cannot accommodate due to limited physical space. Thinking about the owner–dog relationship again, the most important factor deliberated was the cost of separation. Not just because this loss could be profound [6], but because losing their dog could also lead to a quicker deterioration of health. Given that the health of older adults without dogs has been found to be related to faster deterioration of health, it is reasonable to consider that the compounding issue of grief could lead to further health deterioration [15].

The third part of the study’s aim Is predictably the most in-depth; considering what would be the best practice to allow for an effective transition and continued support within the aged care facility. However, it is arguably the most crucial, as it has the potential to guide best practices for future policies within aged care facilities and to facilitate owners in keeping their assistance dog wherever possible. Further, it became evident during the discussions that these policies should not just look at what to do where the owner is able to keep their dog, but also what needs to be put in place where they cannot. As already mentioned, the first step would be to ensure policies outline objective assessments to assess suitability. Thus, the following will discuss further policies and procedures in the context of where owners have the ability or limited ability to care for their dog, and where they are unable to keep their dog, followed by other key considerations.

Where owners are allowed to keep their dogs, it is important to consider the ongoing needs, safety, and welfare of the dog and owner, as well as what to do when unexpected factors arise. It should also be considered that even when the owner is able to care for their dog it may not provide the same utility. Thus, there may be the possibility for retraining (e.g., retraining a hearing dog to alert the owner to a new set of sounds). Nevertheless, key points raised and deliberated were the need for a general animal care plan, and an animal risk management and emergency directive. The general animal care plan should be developed in conjunction with the owner, to establish services needed and any care that should be undertaken by the owner. Where the owner is fully capable, this should be a relatively simple process that can be reviewed periodically as needed, or where an unexpected event occurs (e.g., health deterioration). Where the owner has limited capacity to care for their dog, this plan could include additional assistance depending on their needs, and who will provide that assistance (e.g., a professional or friends/family). This may also include who to contact where temporary respite is needed. It was raised that some aged care facilities have additional services available to residents on a pay-per-use basis. Interestingly, the majority of participants agreed that the provision of dog care support (e.g., walking, feeding) should be a pay-for-use service in an aged care facility. Given that many owners may require assistance, if only periodically, and the limited funding abilities of many aged care facilities, this could be a simple answer to provide a necessary service at no extra cost to the facility. Alternatively, it was raised that there were three other possibilities, including the recruitment of volunteers, which would be no extra cost, the employment of an animal care staff member (particularly where there is a significant number of dogs in the facility), or additional training to current staff to provide dog care. An animal risk management and emergency directive should also be included as part of policy, again developed in conjunction with the owner. This would prevent any confusion over where the dog should be relocated, whether temporarily or permanently, in the case of illness, hospitalisation, or death. This would not only provide peace of mind to staff, but also to owners.

An overlooked factor raised in the discussions was what to do where owners are assessed as unable to keep their dog. We know that this is often associated with a period of grief, loss, and distress [6]. As such, there should be policy protocols in place not only for those facing separation from their assistance dog, but even those facing separation from a companion animal. This could include a transition through decreasing their time with the dog, which would also allow alternatives to be explored and practiced (e.g., cane use). Alternately, having their dog visiting periodically (where possible, such as where family adopt the dog), or having an aged care visiting companion or therapy dog that they are encouraged to engage with. The latter might be a positive decision for the broader facility, with many benefits highlighted in the literature for dog assisted therapy in aged care [16].

Much of the focus thus far has been on the owner and dog, but it is crucial that policies and procedures should also take into consideration others within the aged care facility, such as staff, other residents, and visitors. A myriad of reasons was raised as to why others may not want to be around dogs, such as allergies or fears, so physical separation, signage, training protocols and assistance dog information would be important. Also, some visitors may bring their own dog to the facility, which could interfere with an assistance dog. This also ties into the point that housing and restraint should be considered as part of policy, such as ensuring the dog is kept in the owner’s room where possible, and always leashed when outside the room. This was also raised in the discussion of other factors that should be considered by aged care facilities. Namely, where many dogs are in a facility a dedicated dog-friendly floor or wing could be considered, and rooms should be configured to allow staff movement while the dog is restrained. Though it must be acknowledged that for some facilities there is just not sufficient space to maintain a dog safely or comfortably. This should thus be a key consideration as part of the initial assessment. Ideally, new aged care facilities should be built with animals in mind, and older facilities could consider retrofitting (e.g., adding more access to outdoor spaces). The latter of which would highly benefit from government support, with the provision of grants, particularly to encourage facilities with limited budgets. This was unsurprisingly noted as the most important factor when considering physical space. It was also raised that the dog must be kept at a hygienic standard to manage any risks of injury (such as from long claws) or infection. This should be tied back into the policies and procedures in the dog care plan, whereby the dog receives sufficient ongoing care.

Where staff have identified a sufficient reason why they cannot be around a dog (such as allergies), other staff should be able to work with the owner instead. However, for all other staff, policies should inform sufficient training, whether mandatory or optional, on assistance dogs. It was further suggested this could include staff opting to become a “champion” for assistance dogs, with specific identification like a paw emblem on their name badge. While a very different cause, the use of this idea has previously been successful in the LGBT+ space, with the use of ally training and identification (such as rainbow badges) helping provide a safe, accepting space with the spreading of education [17]. Thus, this may similarly work to encourage acceptance and support of assistance dogs in the facility and further spread education about assistance dog use and etiquette to other residents and visitors.

Given all this, the obvious question that remains is “which relevant bodies should be involved in ensuring these policies are upheld?”. In order of importance from the data, the answer is aged care bodies, assistance dog organisations, and dog rights organisations. This is a logical order, particularly given their relative direct involvement already. But it is important to note that all should be involved to a degree. Aged care bodies would already be involved at a policy level working with aged care facilities, and thus would likely assist in policy implementation support for this case. While assistance dog organisations may not be as involved in the policies, they would often be already in contact with the owner and could be instrumental in the ongoing assessment of the owner’s ability to care for the dog, the necessity of the dog (may no longer have practical utility), and the welfare of the dog. The latter leads to dog rights organisations, which may be required where the welfare of the dog is not assessed by an assistance dog organisation, or where the dog needs to be rehomed.

## 5. Conclusions

The question of whether older adult assistance dog owners should be able to keep their dog on relocation to an aged care facility is a deceivingly complex one. However, by inputting sufficient objective assessments in place for the person, dog, and facility, there is an opportunity to make a logical decision based on the best interests of all involved. Where possible, owner and dog should be kept together, as the law supports, though it is important to have periodic follow-up assessments, care plans, and sufficient policies and procedures to ensure continued safety and welfare.

Policies need to take into consideration the necessity for case-by-case decision making, but also be clear in what should be done where an owner can or cannot keep their dog. And where they can, ensure that this guides procedures that maximise the safety, health, and well-being of all involved, not just the owner and dog, but staff, visitors and beyond. Any facilities that are willing to implement these policies and procedures, thus allowing owners and their dogs to remain together, would be at the forefront of a better future for aged care.

## Figures and Tables

**Figure 1 animals-13-02680-f001:**
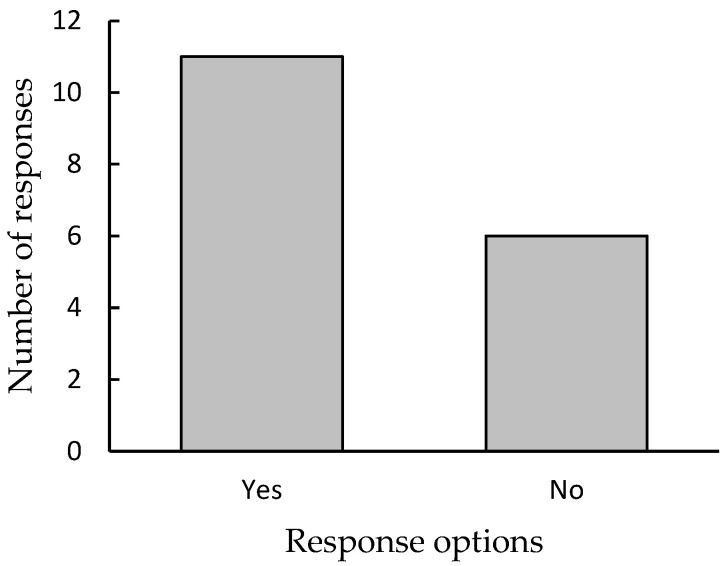
Summary of responses for additional question “Should the provision of dog care support (e.g., walking, feeding) be a pay-for-use service in an aged care facility?”.

**Table 1 animals-13-02680-t001:** Relevant position or background of participants.

Participant No.	Group No.	Position/Background
1	3	Guide dog facilitator and dog trainer; worked in aged care; and former professor of nursing
2	1	Representative from Vision Australia
3	2	Hearing dog user
4	3	Representative from an aged care facility
5	3	Geropsychologist
6	1	Animal therapy director; audiologist
7	2	Representative from the Royal Society of Prevention of Cruelty to Animals (RSPCA)
8	3	Representative from the UQ Business School
9	1	Veterinarian with an interest in older adults
10	2	Veterinarian with an interest in older adults
11	1	Assistance dog trainer
12	3	Researcher in companion dogs in aged care
13	2	Organisational psychologist with an interest in aged care
14	1	Hearing in nursing homes
15	2	RSPCA researcher
16	2	RSPCA dog adoption specialist
17	3	Occupational therapist with a PhD in dementia
18	1	Psychology student and nursing home services consumer

**Table 2 animals-13-02680-t002:** Summary of ranked response statistics by importance for Question 3.

Sub-Response	Importance Rank	M (SD)
Physical environment	First	2.00 (1.03)
	Second	2.27 (1.00)
	Third	2.33 (0.87)
	Fourth	3.40 (1.02)
Staff and training	First	1.88 (0.83)
	Second	2.00 (0.84)
	Third	2.12 (0.76)

**Table 3 animals-13-02680-t003:** Summary of ranked response statistics by importance for Question 4.

Importance Rank	M (SD)
First	1.77 (0.58)
Second	1.85 (0.86)
Third	2.38 (0.84)

**Table 4 animals-13-02680-t004:** Summary of ranked response statistics by importance for Question 5.

Sub-Response	Importance Rank	M (SD)
Person factors	First	2.20 (1.28)
	Second	2.47 (0.88)
	Third	2.60 (1.20)
	Fourth	2.73 (1.00)
Dog factors	First	1.82 (1.04)
	Second	2.53 (1.33)
	Third	3.35 (1.33)
	Fourth	3.59 (1.14)
	Fifth	3.71 (1.23)
Owner–dog relationship factors	First	1.40 (0.49)
	Second	1.60 (0.49)

## Data Availability

Not applicable.

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
