# Peer review of "Cultivating the Acceptance of Assistance Dogs in Aged Care through Deliberative Democracy"

_animals, 2023, doi:10.3390/ani13162680_

Round 1

Reviewer 1 Report

The manuscript is very well written and the methodological approach, i.e. focus group based on deliberative democracy, is in accordance with the objective of the study.

The topic is a very relevant one in terms of societal needs in the context of human-animal interactions with instrumental values for human quality of life. Specifically, the aim of the current study was to explore whether older adults should be allowed to retain  their dog on relocation to an aged care facility, and what factors should impact this decision.

My recommendation is "acceptance with minor revision".

My suggestions for the authors are listed bellow:

- the authors should include in the Introduction part some references to the "One Health, One Welfare" approach, due to the fact that some of their questions that are included in the focus group are covering this approach. 

- in the Methods part: Can the authors specify the composition of the three focus groups? For example, they could include a column in Table 1, in which they show the number of the focus group for each of the participants. Also, in the lines 100-103, the authors indicate that there were three focus groups with a total number of 17 stakeholders included, but in the Table 1 there are 18 participants.

- in the Results sections: I suggest the inclusion of quotes (statements collected from the participants) in order to better reflect the identified themes and sub-themes. 

Author Response

We thank the reviewer for their comments. Please refer to the attached word document for all responses.

Reviewer 2 Report

The paper is overall interesting, but the methodology/design need significant work and much detail.  See notes below. 

Spell out RSPCA in Table 1

Design section needs more detail to explain methodology. Does it generally include qualitative and quantitative data collection? Why was deliberative democracy the best design? Need references for this model also documenting when/how it’s been used.  

Were focus groups in person/online? Who conducted them? What were the “key assumptions” the participants were provided with and what kind of background information?

Line 142 - Author states: “For Questions 1 and 2, sufficient deliberation and agreement was achieved during the focus groups. 143 For Question 3, two key subthemes were identified and included in the questionnaire as 144 two independent answer lists for ranking by importance.”

It’s not clear how this was established.  What determines “sufficient deliberation”? Perhaps if the authors explain this design better at the beginning, this will be answered.

Line 147 - Analysis section – States “Subsequent participant data from this questionnaire was then analyzed using the Qualtrics output” What data was this? Please explain.

Line 267, under Physical Environment – What is meant by “first important response?” What does “important” mean?

Table 2 – would help to have actual wording of response (maybe abbreviated) under Importance Rank, rather than just 1st, 2nd 3rd.

Line 294 – 3.3.3 Where did the additional question come from?

Table 3  & 4 – same suggestion as Table 2

Line 310 – e.g. should be put in parentheses with small letters, for example – Aged care bodies (e.g. aged and community…..)

Author Response

(The authors gave the same response as above.)

Round 2

Reviewer 2 Report

The revised manuscript is greatly improved and I recommend it for publication.